# Functional Analysis of Zebrafish *socs4a*: Impacts on the Notochord and Sensory Function

**DOI:** 10.3390/brainsci12020241

**Published:** 2022-02-10

**Authors:** Monique Trengove, Ruby Wyett, Clifford Liongue, Alister C. Ward

**Affiliations:** 1School of Medicine, Deakin University, Geelong, VIC 3216, Australia; m.trengove@deakin.edu.au (M.T.); rubywyett@gmail.com (R.W.); c.liongue@deakin.edu.au (C.L.); 2Institute for Mental and Physical Health and Clinical Translation, Deakin University, Geelong, VIC 3216, Australia

**Keywords:** SOCS4, SOCS, sensory neuron, notochord

## Abstract

The suppressor of cytokine signaling (SOCS) proteins play important roles in cytokine and growth factor signaling, where they act principally as negative feedback regulators, particularly of the downstream signal transducer and activator of transcription (STAT) transcription factors. This critical mode of regulation impacts on both development and homeostasis. However, understanding of the function of SOCS4 remains limited. To address this, we investigated one of the zebrafish *SOCS4* paralogues, *socs4a*, analyzing its expression and the consequences of its ablation. The *socs4a* gene had a dynamic expression profile during zebrafish embryogenesis, with initial ubiquitous expression becoming restricted to sensory ganglion within the developing nervous system. The knockdown of zebrafish *socs4a* revealed novel roles in notochord development, as well as the formation of a functional sensory system.

## 1. Introduction

The suppressor of cytokine signaling (SOCS) family of proteins have been demonstrated to act as classical negative feedback regulators of cytokine and growth factor signaling. Central to this is the induction of SOCS genes by downstream signal transducer and activator of transcription (STAT) proteins, the activity of which is impacted by SOCS proteins [1]. All SOCS contain a central SH2 domain imparting target specificity, a C-terminal SOCS box involved in the recruitment of an E3 ubiquitin ligase complex, and a more variable N-terminal domain [2]. SOCS1–3, along with the differently named SOCS family member cytokine-inducible SH2-containing protein (CISH), function mainly in cytokine receptor signaling, and have been shown to play particularly key roles in regulating blood and immune cell development and function [3].

Much less is known about SOCS4–7, which have been predominantly implicated in growth factor receptor signaling [4]. Amongst these, SOCS4 remains one of the least studied. The gene encoding SOCS4 has been shown to be widely expressed [5,6], and induced by epidermal growth factor (EGF) [7], while infection can indirectly increase SOCS4 protein levels [8]. Mice carrying a *Socs4* gene mutation are fertile with no overt phenotype, although they succumbed more rapidly to the influenza virus infection [6], while another report has suggested a role for SOCS4 in the regulation of primordial follicle activation in the ovary [9].

This study sought to further our understanding of SOCS4 using zebrafish, which possess two *SOCS4* paralogous, *socs4a* and *socs4b* [10]. Specifically, we investigated the role of *socs4a* through expression analysis and functional investigation using in vivo gene knockdown in zebrafish embryos. Collectively, this revealed key roles for *socs4a* during embryogenesis, impacting the development of the notochord and sensory system.

## 2. Materials and Methods

### 2.1. Zebrafish Methods

Zebrafish lines were housed in an aquatic habitats aquarium facility at the Deakin University Upper Animal House using standard husbandry methods. The following morpholinos (Genetools) were injected into 1–8 cell stage embryos, as described [11]: *socs4a* ATG (5′-AGTTTTTGGTCTTCCTCTCAGACAT) and UTR (5′-ATGTTCTCACAGAAGAGTGGTGAAT), standard control (Ctl) 5′-CCTCTTACCTCAGTTACAATTTATA and p53 [12] MO, along with in-vitro-transcribed *socs4a* mRNA lacking the 5′-UTR. Embryos were raised in Petri dishes using egg-water containing 0.003% (*w*/*v*) 1-phenyl-2-thiourea (PTU) from 9 h post fertilization (hpf) to inhibit pigment formation. As required, F-actin was visualized by the staining of fixed embryos with phalloidin, as described [13], and neuromasts using MitoTracker Red CMX Ros (Invitrogen), as described [14]. To measure touch sensitivity and response, embryos were subjected to a light touch with a fine stimulus, just posterior to their otic vesicle [15], or alternatively subjected to light tapping of the Petri dish.

### 2.2. Reverse Transcription–Polymerase Chain Reaction (RT-PCR)

Total RNA was isolated from zebrafish embryos or adult tissues using TRIzol reagent (Invitrogen) following the manufacturer’s guidelines, and RT-PCR performed as described [16], using the following primers: *socs4a* (5′-GGAAACATTCACCACTCT; 5′-TCCGAGTTTTTGGTCTTCC) and *actb* (5′-AACACAACACAGGATCATGGAG; 5′-CATTGCTACACTTGCTTCTTGC).

### 2.3. In Vitro Transcription and Translation

In vitro transcription and translation of *socs4a* mRNA in the presence of MOs was performed as described [17].

### 2.4. In Situ Hybridization

Digoxygenin (DIG)-labelled RNA probes were used for whole-mount in situ hybridization (WISH), as described [11]. Some hybridized embryos were sectioned and subjected to FastRed counterstaining, as described [18]. DIG-labelled probes were used in combination with fluorescein (FLU)-labelled probes for double fluorescence in situ hybridization (dFISH) [19].

### 2.5. Imaging

Embryos were imaged using an SZX-ILLK200 microscope coupled with a DP90 camera using DP Controller software (Olympus), using either stage lighting or red fluorescence as appropriate, or using a Fluorview 1000 scanning confocal microscope and FV1000 Viewer software (Olympus).

## 3. Results

### 3.1. Expression Analysis of Zebrafish socs4a during Embryogenesis

To gain insight into potential roles for *socs4a*, semi-quantitative RT-PCR was performed on staged zebrafish embryos. This identified *socs4a* expression throughout embryogenesis, but with a dynamic expression profile (Figure 1A). Transcripts of *socs4a* were evident at the 1-cell stage, indicating maternal derivation, but these then declined slightly by 12 h post fertilization (hpf), before increasing again by 16 hpf, before decreasing from 31 hpf; however, steady expression remained to 6 days post fertilization (dpf). The expression of *socs4a* was also observed in all adult tissues tested, with highest transcript levels identified in the brain (Figure 1B).

To determine the spatio-temporal expression during embryogenesis, whole-mount in situ hybridization (WISH) with an anti-sense probe against the full-length *socs4a* coding region was employed. Ubiquitous expression was seen from the 1-cell stage (Figure 1C), but by 16 hpf a distinct expression pattern had emerged, with bilateral expression in presumptive cranial ganglion (CG), the paired precursor dorsal root ganglion (pDRG) in the trunk, and in neurons within the developing tail (Figure 1D,E). By 24 hpf, the expression domain had further expanded, with bilateral expression noted in several sensory cranial ganglion, including the trigeminal ganglion (TgG), anterior and posterior lateral line ganglion (ALLG, PLLG), statoacoustic ganglion (SAG), and the fifth (V), sixth (VI) and seventh (VII) cranial nerves, while expression in the pDRG also persisted (Figure 1G–I). Expression in the majority of these structures waned from 48 hpf (Figure 1J), with expression remaining only in the ALLG and PLLG by 7 dpf (Figure 1K). A *socs4a* sense probe was used at as a negative control, and failed to produce significant staining (Figure 1F and Appendix A).

The discrete bilateral expression pattern of *socs4a* was consistent with the expression in developing neuronal structures. To confirm their identity, double fluorescent in situ hybridization (dFISH) was employed using both a probe targeting *socs4a,* and one targeting a neuronal marker. At 24 hpf, clear co-expression was seen between *socs4a* and the pan-neuronal marker *elavl3* [20] in a subset of the neuronal cells within the PLLG, with independent expression elsewhere (Figure 1M). Similar analysis with the differentiating neuronal marker *neurod1* [21] showed limited co-expression (Figure 1N). In contrast, dFISH analysis with the sensory and motor neuronal specific marker *isl1a* [22] revealed co-expression in the TgG, ALLG and cranial nerve V within the hindbrain (Figure 1O). In contrast, the expression of *socs4a* did not overlap with the *isl1a+* sensory motor neurons of the spinal cord (Figure 1P).

### 3.2. Targeted Knockdown of Zebrafish socs4a

To examine the function of zebrafish *socs4a* during embryogenesis, anti-sense morpholino (MO)-mediated gene knockdown [23] was employed, independently targeting the start codon (ATG MO) and a region of the 5′UTR (UTR MO) (Figure 2A). The efficacy of the ATG MO was confirmed using in vitro transcription–translation of *socs4a* performed in the presence of either the ATG MO or a control (Ctl) MO. A protein product of the appropriate size for Socs4a (44 kD) was generated with the Ctl MO, which was substantially abrogated in the presence of the ATG MO (Figure 2B) and absent without the *socs4a* mRNA template (Appendix A).

Embryos injected with the *socs4a* ATG MO demonstrated a ventral tail curl from 24 hpf (Figure 2D,E), which continued with high penetrance (82.4%) to 48 hpf (Figure 2J) and beyond (data not shown). The UTR MO also produced a ventral tail curl, although at a reduced penetrance (36.1%) (Figure 2G,J), whereas embryos treated with Ctl MO did not display any curvature phenotypes (Figure 2C,F,J). To reinforce the specificity of the ventral curl phenotype, the UTR MO was co-injected with *socs4a* mRNA lacking the 5′UTR, which produced a partial rescue (Figure 2H,J). In contrast, co-injection of a p53 MO to abrogate potential non-specific cell death [12] did not significantly impact the phenotype observed (Figure 2I,J).

In order to elucidate potential mechanisms underlying the ventral tail curl, light microscopy revealed the notochord, a rod-shaped structure defining the primitive anterior–posterior (*A-P*) axis and essential for providing support, was disrupted in *socs4a* morphant embryos, a phenotype which varied in severity. Close examination of the large vacuolated epithelial cells that fill the notochord revealed no obvious morphological abnormalities, although cells appeared to be slightly compressed when compared with the control embryos (Figure 2K,L). Defects in the floor plate, the ventral most component of the neural tube that runs parallel with the notochord, have been associated with notochord curvature in zebrafish [24]. However, the floor plate along with the hypochord were clearly present in morphants (Figure 2K,L). Measurement of morphant embryos at 4 dpf revealed that those displaying notochord phenotypes were significantly shorter along the *A-P* axis than age-matched control embryos, including at 48 hpf (Figure 2M). Light microscopic examination also revealed that somite boundaries in *socs4a* morphants were often not linear, but instead were curved (Figure 2N,O). Muscles fibers of the somites were labeled with phalloidin to highlight F-actin, which largely confirmed results seen by light microscopy, with individual muscle fibers structurally normal, but with a disturbed distribution as a result of overall ventral curvature (Figure 2P,Q). This is a likely consequence of inadequate structural support from the notochord [24].

### 3.3. Molecular Analysis of socs4a Knockdown

To more closely examine notochord and floor plate integrity, morphants were subjected to WISH with *sonic hedgehog a* (*shha*), a marker of the developing notochord and floor plate [25]. This revealed severe undulations of the notochord at 24 hpf in *socs4a* morphants (Figure 3B,E) and even split posterior notochords in a small number of embryos (Figure 3C), absent in controls (Figure 3A,D), although floor plates were present and the level of expression appeared unaffected. Morphants were also examined for expression of *ihhb*, present in the chordamesoderm before its differentiation into notochord and prior to inflation of the vacuoles [26]. This was expressed at normal levels at 24 hpf in *socs4a* morphants (Figure 3G), compared with the controls (Figure 3F), but again in an undulating pattern. Staining with *shha* at 10 hpf (Figure 3H–J) revealed the undulations and split notochords were already present at this early time-point.

To further investigate the phenotype observed, the cells of the dorsal midline (the presumptive cells of the notochord) were examined during the processes of convergence and extension. The *T box transcription factor Ta* (*tbxta*) gene is expressed in the shield and presumptive notochord cells, and is essential for the development of the notochord and the convergence process [27]. Expression levels of *tbxta* in the germ ring and shield of *socs4a* morphants were relatively unaltered at 80% epiboly, but the distribution of *tbxta+* cells was altered. Indeed, in a number of *socs4a* morphants embryos, *tbxta+* cells did not completely converge at the midline and did not fully extend from the vegetal pole together towards the animal pole (Figure 3K–M). To confirm this potential convergence and extension defect, the expression of *notochord homeobox* (*noto*) (Figure 3N–P), a gene which marks cells fated to become notochord [24], and *snail1a* (*snai1a*) (Figure 3Q–S), a marker for involuting cells of the germ ring adjacent to the dorsal midline [28], were examined. In both cases, incomplete convergence of the marked cells was revealed.

To determine whether this phenotype also affected tissues surrounding the notochord, *myogenic differentiation 1* (*myod1*) was used to stain paraxial mesoderm, which flanks the developing notochord and neural tube and gives rise to somites, and ultimately muscle [29]. The expression of *myod1* in the paraxial mesoderm of *socs4a* morphants at 9 somites was again undulated, with the expression in the developing somites reduced (Figure 3T,U). The distance between the paired paraxial mesoderm tissues was also found to be significantly greater in morphants at 9 somites (Figure 3T–V), with the distance from anterior to posterior measured across the yolk, also significantly longer (Figure 3W–Y). These data confirm *socs4a* morphants were significantly shorter and wider than the control embryos, consistent with disrupted convergence/extension.

Notochord signaling also plays a role in zebrafish pancreas patterning, with areas of *shha* signaling associated with an absence of the expression of pancreatic markers [30]. Therefore, the expression of *insulin* (*ins*), a marker of the endocrine pancreas-specific β-cells [31] was examined. At 30 hpf, ectopic *ins* expression was noted in *socs4a* morphants, with 43.8% of *socs4a* ATG morphants displaying at least two distinct populations of *ins+* cells (Figure 3Z,A’).

### 3.4. Effect of socs4a Knockdown on the Sensory System

Expression of *socs4a* was identified in several cranial nerves, including both cranial ganglion and presumptive dorsal root ganglion, which contain cell bodies of afferent sensory nerves from the anterior and trunk, respectively. WISH analysis revealed substantially reduced *isl1a* expression in the sensory motor neurons of the hindbrain and anterior portion of the trunk from 26 hpf in *socs4a* morphants, compared with the controls (Figure 4A–C’), while a minor change in the pattern of *neurod1* expression was also observed (Figure 4D,E). Given these observations, the mechanosensory response was investigated. Touch sensitivity was found to be significantly reduced in both *socs4a* morphants compared with the controls (Figure 4F). Expression of *socs4a* was also observed in the anterior and posterior lateral line ganglion, which innervate the neuromast structures of the lateral line, which is responsible for the detection of water movement [32]. However, *socs4a* morphants showed no difference in the number, distribution, or gross morphology of neuromasts compared with the controls (Figure 4G,H). The function of the lateral line neuromasts was tested by an analysis of vibration detection and response at 3 dpf, when the lateral line mechanosensory system is fully functional. No significant difference in response to water vibration was observed with *socs4a* morphants when compared to the controls (Figure 4I).

## 4. Discussion

SOCS4 remains one of the least well-characterized members of the SOCS family, with its expression yet to be fully described, and only limited functional studies. Utilizing the zebrafish as a powerful alternative developmental model, this study aimed to elucidate the embryonic expression profile and in vivo function of the zebrafish *socs4a* paralogue.

Zebrafish *socs4a* was found to be expressed throughout embryonic development, with the highest transcript levels identified from 16–48 hpf, a critical period in neuronal development [33]. WISH analysis identified a dynamic and complex spatio-temporal expression pattern during embryogenesis. Transcripts were observed in 1-cell embryos, confirming maternal derivation, as zygotic expression does not begin until at least 6 hpf [34], which remained broad during gastrulation and epiboly. However, by 16 hpf, *socs4a* expression had become restricted to developing cranial ganglion and peripheral neurons, and by 24 hpf expression had expanded to include several more cranial ganglion and neurons, associated with touch sensation, and the anterior and posterior lateral line ganglion, associated with the innervation of the neuromasts in the lateral line. The largely neuronal embryonic expression pattern was consistent with analysis of adult zebrafish tissue, with the highest *socs4a* expression seen in the brain. A study of adult zebrafish habenular neuronal cell types has also documented *socs4a* expression in dorsal habenular, especially with a *cbln2b*+ cluster associated with sensory response [35]. With the exception of the olfactory bulb, mammalian Socs4 expression has not been described in neuronal tissues but rather thymus and intestine [5,6].

To investigate *socs4a* function, a morpholino-mediated knockdown strategy was employed. Analysis with *noto* and *tbxta*, genes essential for convergence/extension and notochord development, indicated the reduced convergence of cells medially, and the reduced extension of cells towards the animal pole in *socs4a* morphants. This was confirmed with *snai1a*, a marker for cells located outside the shield, as well as a general disorganization of cells at these sites. Correct convergence extension is essential for the normal development of the embryonic notochord. Analysis with *shha* revealed that the notochord was disrupted from its normal rod-like appearance. In some rare cases, a “split” notochord phenotype was observed, likely due to the incomplete convergence of cells fated to develop the notochord. However, analysis using both light microscopy and WISH analysis with *shha* and *ihhb* revealed that the notochord, floor plate and hypochord were all present. These data, taken together with the convergence extension defects observed, suggest that failure of the notochord to converge medially and elongate fully is not an intrinsic notochord defect *per se*, but instead a secondary effect. The notochord provides structural support, as well as secreting signaling molecules influencing surrounding tissues [24]. Analysis of *myod1* expression revealed the paraxial mesoderm and developing somites, which lie adjacent to the notochord, also had an undulating pattern in *socs4a* morphants, as well as reduced *myod1* expression. Light microscopy and phalloidin staining of muscle fibers revealed undulating and curved somite boundaries in morphants, while also revealing individual muscle fibers were less extended horizontally than those seen in control embryos, concomitant with a decrease in overall body length. Perturbations to somite and muscle development are likely an indirect consequence of abnormal convergence extension and/or incomplete notochord elongation. Signaling by *shha* from the notochord also plays a role in the expression of *ins* in the β-cells of the endocrine pancreas [36]. In contrast to the single population of *ins*+ cells, located medially, a large portion of *socs4a* morphants had multiple distinct *ins+* cell populations, suggesting that the abnormal notochord may indirectly affect patterning. Alternatively, convergence of the two endocrine precursor populations medially may have been incomplete in morphants as part of the convergent extension defect, resulting in additional *ins*+ cell populations.

Analysis with the *isl1a* marker, expressed in many cranial ganglion and sensory neurons, revealed a significant reduction of *isl1a*+ cells in the hindbrain and anterior trunk at 22 hpf. It is hypothesized that these cells were likely presumptive dorsal root ganglion or Rohon–Beard neurons, both of which are involved in mediating mechanosensory sensation. Reduced *isl1a+* cells were confirmed at 26 hpf, which suggests that these sensory cells failed to develop in *socs4a* morphants, rather than simply being delayed in development. The *socs4a* morphants were also found to be less responsive to touch. It is yet to be confirmed whether the decrease in *isl1a* expression or reduction in size/function of these sensory neurons is directly linked to the touch response defect identified in *socs4a* morphants. However, a similar correlation between reduced *isl1**a+* motor neuron in the hindbrain and perturbed touch response was observed in *exosc8* morphants [37]. The mechanism behind this loss of *isl1a* expression is yet to be fully investigated. Finally, the strong expression of *socs4a* in the anterior and posterior lateral line ganglion were examined further. Neuromasts of the lateral line, the sensory organs responsible for detection of water movement, are innervated by the lateral line ganglion [38]. However, close examination of neuromasts by Mitotraker Red staining did not reveal any structural abnormalities in *socs4a* morphants, and there was no detectable difference in water vibration response between *socs4a* morphants and controls.

Zebrafish possess two *socs4* paralogues [10]. Analysis of other gene paralogues have shown that such genes may retain an overlapping function to their mammalian orthologues, or may have divided the functional roles of the original orthologue, or one or both of the genes may have taken on a novel function [39]. The encoded socs4a protein is more divergent from human SOCS4 than the socs4b protein [10], especially in the SH2 domain that mediates target specificity [3]. In addition, there was no conservation of the *let-7* microRNA binding site found in the 3′UTR of mammalian *SOCS4* genes in zebrafish *socs4a* (data not shown), indicating that this mode of regulation is not conserved for *socs4a*. Moreover, the strong neuronal expression of *socs4a* differs from that observed with the *SOCS4* gene. Collectively, this suggests *socs4a* may have diverged in function compared with mammalian *SOCS4*. The generation and characterization of germline *socs4a* mutants would provide confirmation of this during embryogenesis, and enable further studies into adulthood.

LIF activation through the JAK1/STAT3 pathway has been shown to induce SOCS4 expression during ovarian follicular development [9], consistent with the strong levels of maternally derived *socs4a* transcripts observed. Moreover, the expression of zebrafish *stat3* overlaps strongly with that of *socs4a* during embryogenesis, including in a variety of sensory neural structures, such as the ALLG, PLLG, TG and SAG [40,41]. Interestingly, zebrafish *stat3* has been implicated in convergence/extension [40], with *stat3* mutants displaying reduced A-P extension [42], as observed in *socs4a* morphants. However, mammalian SOCS4 has been shown to inhibit STAT3 activation [43]. Indeed, in vivo studies have suggested that this may represent a tumor suppressor role, with RUNX1 able to directly repress *Socs4* expression leading to increased STAT3 activity, which contributes to tumor development [44]. So, whether SOCS4 functions as an inducible negative feedback regulator of STAT3, similar to CISH and STAT5 [17], or instead lies downstream of STAT3, remains entirely speculative.

SOCS4 expression is also modulated by infection [8], and has also been shown to increase virus clearance and protect against severe cytokine storm during influenza infection [6]. It will be of interest to pursue infection studies in *socs4a* morphants to understand whether this function is conserved. In addition, the functional analysis of zebrafish *socs4b* will undoubtedly produce further insights into SOCS4 function.

## Figures and Tables

**Figure 1 brainsci-12-00241-f001:**
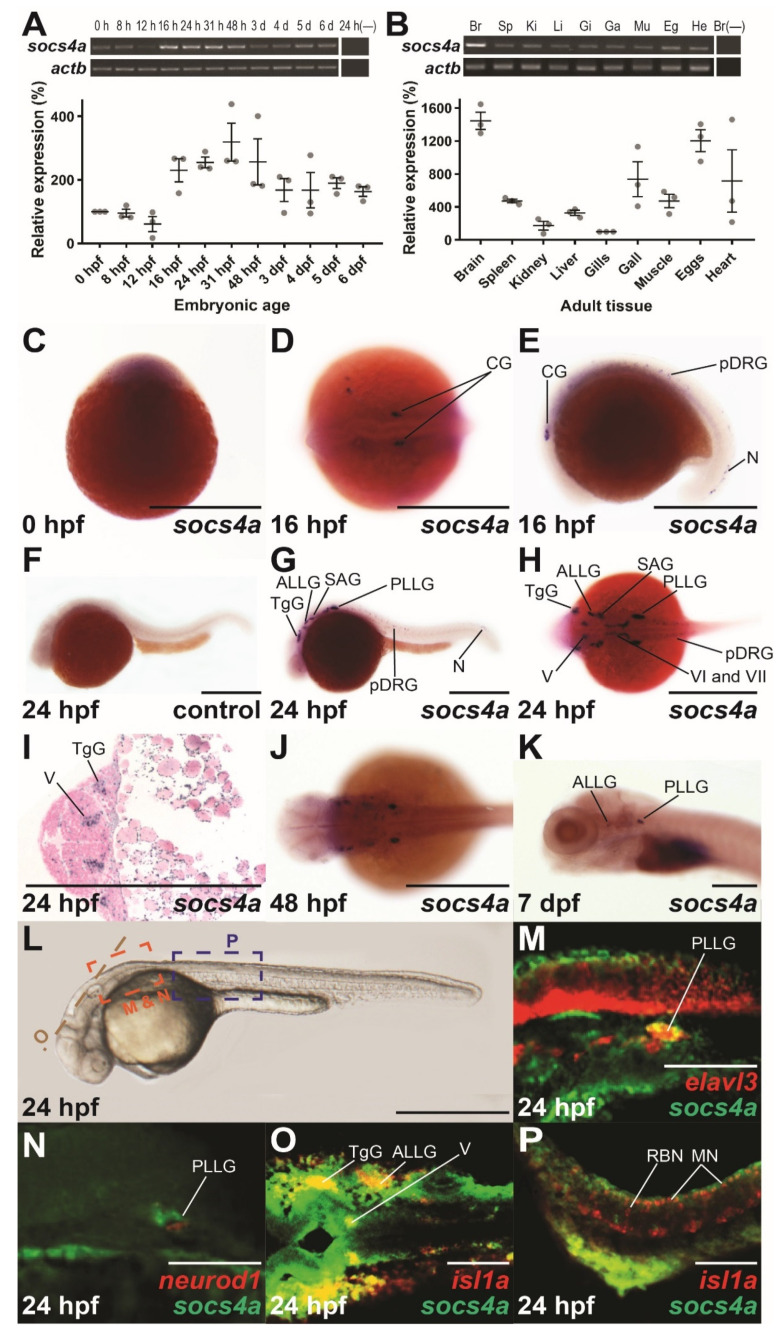
Spatio-temporal expression profile of *socs4a*. RT-PCR analysis of *socs4a* and control *actb* (β-actin) from total RNA extracted from embryos at the indicated times post fertilization ((**A**), upper panel) or from the indicated adult tissues ((**B**), upper panel), with a no RT control included (−). Levels of *socs4a* were quantified using densitometry and standardized to the *actb* gene with the embryonic expression profile shown relative to expression at 0 hpf ((**A**), lower panel) and the adult profile relative to expression in the gills ((**B**), lower panel). Whole-mount in situ hybridization analysis of *socs4a* on embryos at 0 hpf (**C**), 16 hpf (**D**,**E**), 24 hpf (**F**–**I**), 48 hpf (**J**), and 7 dpf (**K**) using either sense (control) or anti-sense (*socs4a*) probes, as indicated. Expression is demonstrated by the presence of blue/purple staining. dFISH on 24 hpf embryos with anti-sense probes for *socs4a* and either *elavl3* (**M**), *neurod1* (**N**) or *isl1a* (**O**,**P**), with the regions of the embryo imaged displayed in panel **L**. Expression is demonstrated by red or green fluorescence for each marker, as indicated, with areas of co-expression being yellow. The embryo in panel C is upright and imaged laterally. All other embryos are positioned with their anterior to the left and imaged laterally (**E**–**G**,**K**–**N**,**P**), dorsally (**D**,**H**,**J**,**O**) or represent a cross-section (**I**). Scale bars = 0.5 mm (**C**–**L**) or 0.1 mm (**M**–**P**). Abbreviations: ALLG, anterior lateral line ganglion; Br, brain; CG, cranial ganglion; Eg, eggs; Ga, gall bladder; Gi, gills; He, heart; Ki, kidney; Li, liver; MN, motor neuron; Mu, muscle; N, neuron; pDRG, precursor dorsal root ganglion; PLLG, posterior lateral line ganglion; RBN, Rohan–Beard neuron; SAG, statoacoustic ganglion; Sp, spleen; TgG, trigeminal ganglion; V, fifth cranial nerve; VI, sixth cranial nerve; VII, seventh cranial nerve.

**Figure 2 brainsci-12-00241-f002:**
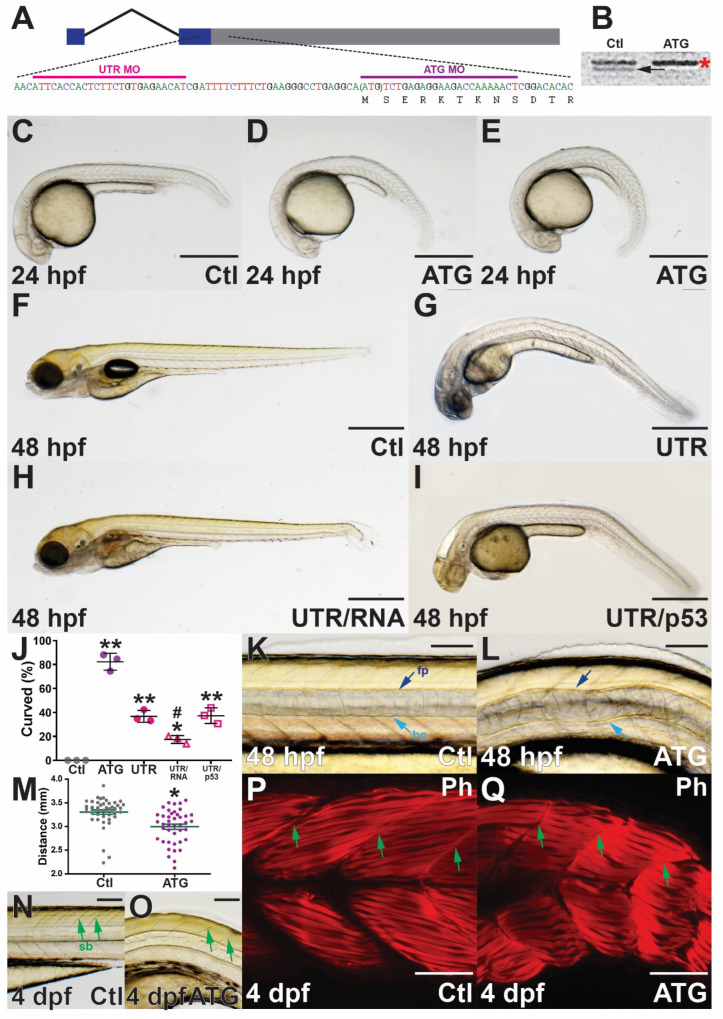
Phenotypic consequences of *socs4a* knockdown. Schematic of *socs4a* mRNA schematic showing exons boxed and joined by a thin line, with non-coding sequence in blue and coding in grey. Sequences targeted by ATG MO (purple) and UTR MO (pink) are indicated (**A**). Confirmation of ATG MO efficacy by in vitro transcription and translation of *socs4a* in the presence of control (Ctl) or ATG MO with the arrow indicating the position of the Socs4a protein at ~44 kDa and the asterisk a non-specific protein (**B**). Analysis of embryos injected with Ctl (**C**,**F**,**K**,**N**,**P**), ATG (**D**,**E**,**L**,**O**,**Q**) or UTR (**G**) MO, or combined UTR MO with MO-resistant *socs4a* mRNA (UTR/RNA, (**H**)) or with p53 MO (UTR/p53, (**I**)) by either light microscopy (**C**–**I**,**K**,**L**,**N**,**O**) or fluorescence microscopy of phalloidin (Ph) staining of muscle fibers (**P**,**Q**). All embryos are positioned with their anterior to the left and images are lateral view. Scale bars = 0.5 mm (**C**–**I**) or 0.1 mm (**K**,**L**,**N**,**Q**). The ventral curl phenotype in Ctl, ATG, UTR, UTR/RNA and UTR/p53 MO was quantified, and shown as mean ± SEM, with statistical significance versus Ctl (*p* < 0.05: *; *p* < 0.01: **) or UTR (*p* < 0.05: #) shown (**J**). Embryo length (anterior–posterior) at 4 dpf is provided for individual embryos, with mean ± SEM and statistical significance versus Ctl indicated (*p* < 0.05: *) (**M**). Abbreviations: fp, floor plate; hc, hypochord; sb, somite boundary.

**Figure 3 brainsci-12-00241-f003:**
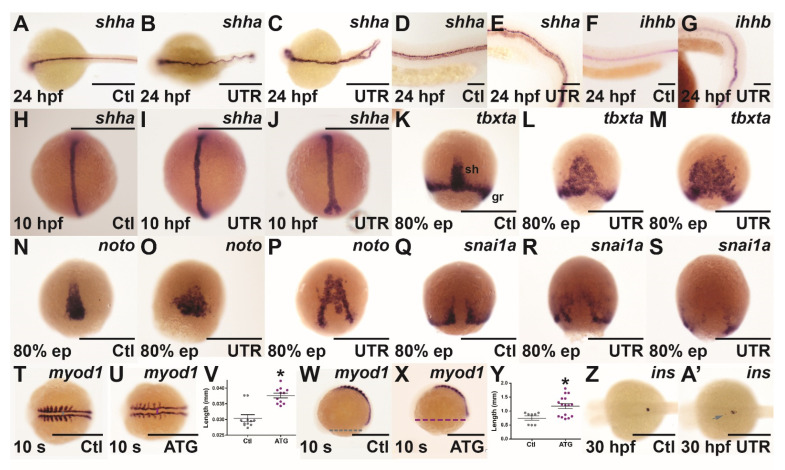
Effect of *socs4a* knockdown on convergence and extension. Embryos injected with either Ctl (**A**,**D**,**F**,**H**,**K**,**N**,**Q**,**T**,**W**,**Z**), UTR (**B**,**C**,**E**,**G**,**I**,**J**,**L**,**M**,**O**,**P**,**R**,**S**,**A’**) or ATG (**U**,**X**) MO were subjected to WISH at 24 hpf (**A**–**G**), 10 hpf (**H**–**J**), 80% epiboly (ep) (**K**–**S**), 10 somites (s) (**T**,**U**,**W**,**X**) and 30 hpf (**Z**,**A’**) for *shha* (**A**–**E**,**H**–**J**), *ihhb* (**F**,**G**), *tbxta* (**K**–**M**), *noto* (**N**–**P**), *snai1a* (**Q**–**S**), *myod1* (**T**,**U**,**W**,**X**) and *ins* (**Z**,**A’**). Both mild (**B**,**I**,**L**,**O**,**R**) and more severe (**C**,**J**,**M**,**P**,**S**) phenotypes are shown. Scale bars = 0.5 mm (**A**–**C**,**H**–**U**,**W**,X,**Z**,**A’**) or 0.1 mm (**D**–**G**). The distance between parallel paraxial mesoderm structures (**V**) and across yolk from anterior to posterior for 10 somite embryos was measured (**Y**). Results are shown for individual embryos, along with mean ± SEM, with statistical significance relative to Ctl indicated (*p* < 0.05: *). Abbreviations: gr, germ ring; sh, shield.

**Figure 4 brainsci-12-00241-f004:**
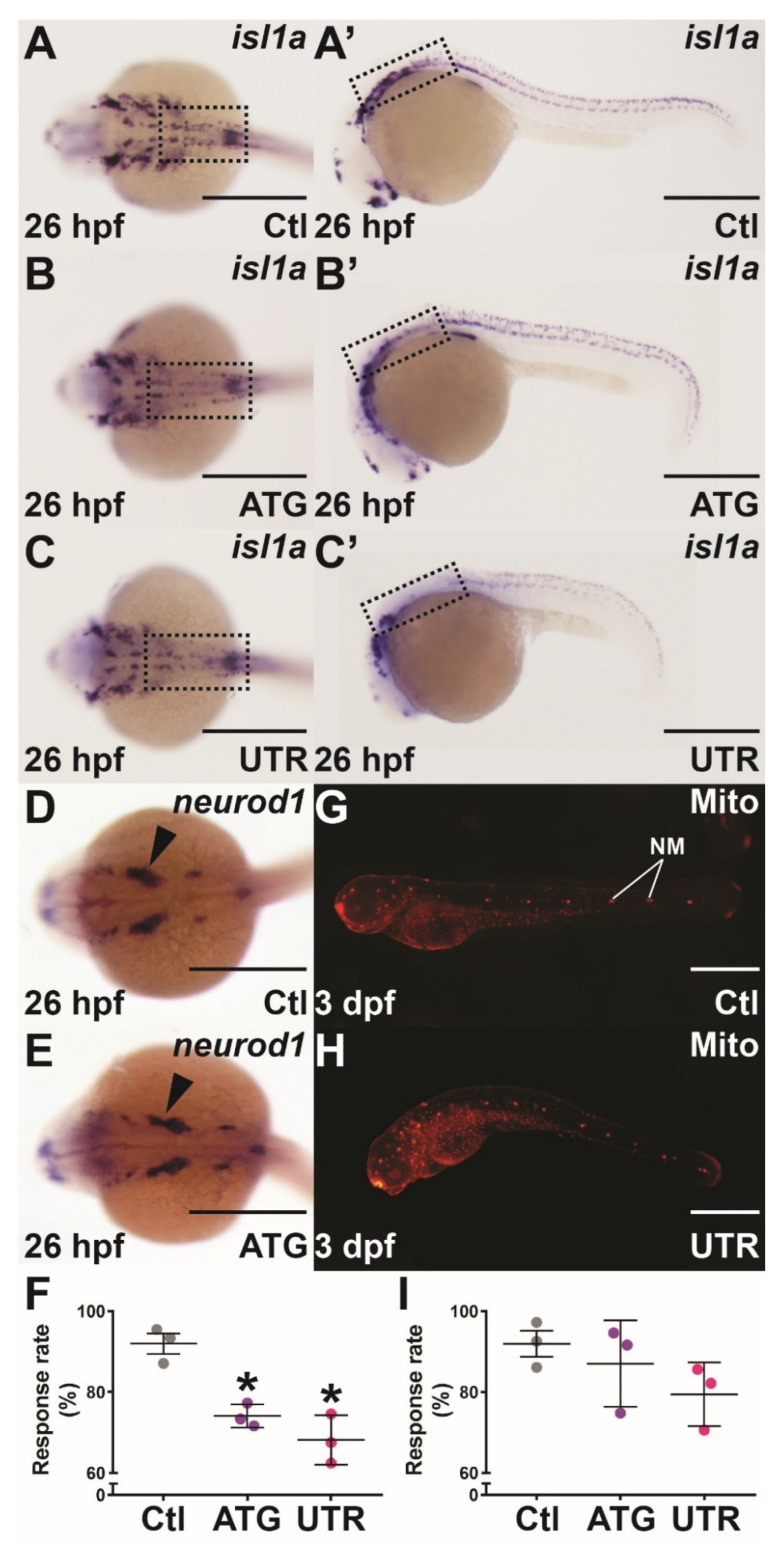
Mechanosensory analysis in *socs4a* knockdown embryos. Embryos injected with Ctl (**A**,**A’**,**D**,**G**), ATG (**B**,**B’**,**E**) or UTR (**C**,**C’**,**H**) MO were subjected to WISH with *isl1a* (**A**–**C’**) or *neurod1* (**D**,**E**) at 26 hpf, or Mitotracker red (Mito) staining at 3 dpf (**G**,**H**) and imaged dorsally (**A**–**E**) and laterally (**A’**–**C’**,**G**,**H**), with the boxed areas highlighting the equivalent areas in the alternative images (**A**–**C**,**A’**–**C’**). Scale bars = 0.5 mm. Alternatively, the responsiveness to touch at 24 hpf (**F**) and water movement at 3 dpf (**I**) was quantified, with results presented for 3 independent experiments (*n* > 30 embryos in each group) along with mean ± SEM and statistical significance relative to Ctl (*p* < 0.05: *). Abbreviation: NM, neuromast.

## Data Availability

All data generated or analyzed during this study are included in this published article (and its supplementary information files).

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
