# Peer review of "Functional Analysis of Zebrafish socs4a: Impacts on the Notochord and Sensory Function"

_brainsci, 2022, doi:10.3390/brainsci12020241_

Round 1
Reviewer 1 Report
The manuscript brainsci-1466099, entitled “Functional analysis of zebrafish socs4a: impacts on the notochord and sensory function”, by Trengove and colleagues, is a report on the expressional profile of the zebrafish socs4a gene, accompanied by a functional analysis, preliminarily performed using morpholino-mediated antisense strategy. The gene knock-down impairs notochord and sensory system formation during zebrafish embryonic development.
Major points
Morpholino experiments are well controlled. However, a preliminary evaluation of socs4a crispant embryos (F0 generation analyzed under CRISPR/Cas-induced transient conditions) could provide a more robust comparison, at least focusing on 1-2 phenotypes like those described in morphant embryos.
To check if a crosstalk between socs4a and Stat3 signaling is in place, the author could treat socs4a morphants with Stat3 agonists or inhibitors, checking if the morphant phenotypes can be rescued or exacerbated by these treatments. In parallel, Stat3-targeted embryos could be analyzed for socs4a expression.
Throughout the whole manuscript, official names for the mentioned genes should be checked, according to the ZFIN database. Data dispersion should be displayed in the bar charts.
Minor points
Introduction
Line 32: please explain the meaning of CISH; the whole sentence appears not very clear; “alternatively” to what? Please complete the missing information.
Methods
Petri should be with capital P. “In situ” should be written in italics.
Results
Scale bars are missing in Figure 1.
Is the isl1 marker corresponding to the isl1a or isl1b gene?
Is there a supplementary figure displaying the sense probe results at all considered stages? This would be relevant to compare the staining of antisense and sense probes, especially at the 1-cell stage.
Maternal derivation could be further checked by RT-PCR analysis on unfertilized eggs.
The new official name for the huc marker is elavl3.
Scale bars are missing in Figure 2.
Line 149: please check for a possible repetition in describing the L panel of Fig. 2.
Could the authors display the entire gel related to panel B of Figure2, including the size marker used to evaluate the samples dimensions?
Line 187: a full stop is missing.
Scale bars are missing in Figure 3.
Is the shh marker corresponding to the shha or shhb gene?
The new name for the ntl marker is tbxta
The new name for the flh marker is noto
The official name for the sna1 marker is snai1a
The official name for the myod marker is myod1
The official name for the neurod marker is neurod1
Line 239: an extra space is present
Scale bars are missing in Figure 4.
The legend in Fig. 4 should describe the dotted areas and the NM abbreviation.
Discussion
Line 255: is maybe a verb missing?
Line 268: associated
Line 326: an extra space is present
Author Response
Reviewer 1:
The manuscript brainsci-1466099, entitled “Functional analysis of zebrafish socs4a: impacts on the notochord and sensory function”, by Trengove and colleagues, is a report on the expressional profile of the zebrafish socs4a gene, accompanied by a functional analysis, preliminarily performed using morpholino-mediated antisense strategy. The gene knock-down impairs notochord and sensory system formation during zebrafish embryonic development.
Major points
Morpholino experiments are well controlled. However, a preliminary evaluation of socs4a crispant embryos (F0 generation analyzed under CRISPR/Cas-induced transient conditions) could provide a more robust comparison, at least focusing on 1-2 phenotypes like those described in morphant embryos.
- The generation and characterization of socs4a ‘crispant’ embryos was not possible in the limited timeframe for submission. However, this excellent suggestion has been added to the Discussion. In addition, a ‘rescue’ experiment has been included (new Fig. 2H, J), which provides further validation of the morpholino experiments.
To check if a crosstalk between socs4a and Stat3 signaling is in place, the author could treat socs4a morphants with Stat3 agonists or inhibitors, checking if the morphant phenotypes can be rescued or exacerbated by these treatments. In parallel, Stat3-targeted embryos could be analyzed for socs4a expression.
- Preliminary experiments with Stat3 inhibitors failed to provide definitive results regarding impacts on the morphant phenotype – complicated by the fact that Stat3 ablation on its own alters A-P axis elongation. Therefore, the statement regarding the possible involvement of Stat3 has been removed from the Abstract and made more speculative in the Discussion, with an additional relevant reference added (new reference #42).
Throughout the whole manuscript, official names for the mentioned genes should be checked, according to the ZFIN database.
- As requested, all gene names have been updated to the official gene names.
Data dispersion should be displayed in the bar charts.
- As requested, the dispersion of individual data points is now displayed on all graphs.
Minor points
Introduction
Line 32: please explain the meaning of CISH; the whole sentence appears not very clear; “alternatively” to what? Please complete the missing information.
- As suggested, the sentence has been reworded, with ‘alternatively’ replaced with ‘differently’ (line 31).
Methods
Petri should be with capital P. “In situ” should be written in italics.
- Both changes made throughout.
Results
Scale bars are missing in Figure 1.
- Scale bars added.
Is the isl1 marker corresponding to the isl1a or isl1b gene?
- The marker used is isl1a, which has been clarified throughout.
Is there a supplementary figure displaying the sense probe results at all considered stages? This would be relevant to compare the staining of antisense and sense probes, especially at the 1-cell stage.
- As suggested, a new Supplementary Figure (Supp. Figure 1) has now been added displaying sense probe results across multiple time points.
Maternal derivation could be further checked by RT-PCR analysis on unfertilized eggs.
- As mentioned in the Discussion, zygotic expression does not commence until several hours after fertilization (reference #34), so the 0 hpf sample only contains maternal transcripts.
The new official name for the huc marker is elavl3.
- The manuscript has been revised to use the new gene name throughout.
Scale bars are missing in Figure 2.
- Scale bars added. In addition, the labelling to panels L, O and Q have been corrected to ‘ATG’ rather than ‘UTR’ MO. We apologise for this error, which does not affect the conclusions drawn.
Line 149: please check for a possible repetition in describing the L panel of Fig. 2.
- The Fig. 2 legend has been revised to remove repetition.
Could the authors display the entire gel related to panel B of Figure2, including the size marker used to evaluate the samples dimensions?
- As suggested, a new Supplementary Figure (Supp. Figure 2) has now been added displaying the entire gel – including size markers – related to Fig. 2B.
Line 187: a full stop is missing.
- Full stop has been added.
Scale bars are missing in Figure 3.
- Scale bars added.
Is the shh marker corresponding to the shha or shhb gene?
- The marker used is shha, which has been clarified throughout.
The new name for the ntl marker is tbxta
The new name for the flh marker is noto
The official name for the sna1 marker is snai1a
The official name for the myod marker is myod1
The official name for the neurod marker is neurod1
- The manuscript has been revised to use the new names of each of these 5 genes throughout.
Line 239: an extra space is present
- Space removed.
Scale bars are missing in Figure 4.
- Scale bars added.
The legend in Fig. 4 should describe the dotted areas and the NM abbreviation.
- As suggested, the dotted area is now described in the Fig. 4 legend. Note that the ‘NM’ (=’neuromast’) abbreviation is already included (below the Figure).
Discussion
Line 255: is maybe a verb missing?
- Appropriate verb added.
Line 268: associated
- Typographical error corrected.
Line 326: an extra space is present
- Space removed.
Reviewer 2 Report
- Figure 2F-H: The control embryos look much smaller than the MO-injected embryos. This seems to be due to photography. More comparable pictures need to be provided.
- Figure 2O-P: The difference in phalloidin staining between control and MO-injected embryos is not clear. Better pictures should be provided.
- The specificity of the morpholinos must be validated. First, the knockdown effect in the morphants needs to be validated at the protein level. Second, a rescue experiment with an mRNA, which encodes for Socs4a but has a modified 5′‐UTR or splice site that is not targeted by the morpholino, should be performed.
- What is the dark region in Figure 1K? Is is part of the experimental staining or background staining?
- Page 7 line 243: There is probably a typo in Fig. 4K, it is likely to be 4I.
- There is no data that supports a potential interaction between socs4a gene and STAT3 signaling. Thus the conclusions that infer such an interaction should be modified accordingly.
Author Response
Reviewer 2:
Figure 2F-H: The control embryos look much smaller than the MO-injected embryos. This seems to be due to photography. More comparable pictures need to be provided.
- As suggested, the images in these panels have been revised to be at the same magnification for easier comparison.
Figure 2O-P: The difference in phalloidin staining between control and MO-injected embryos is not clear. Better pictures should be provided.
- Larger images have been provided to improve clarity (new Fig. 2P-Q).
The specificity of the morpholinos must be validated. First, the knockdown effect in the morphants needs to be validated at the protein level. Second, a rescue experiment with an mRNA, which encodes for Socs4a but has a modified 5′‐UTR or splice site that is not targeted by the morpholino, should be performed.
- As suggested, a rescue experiment has been included in which a socs4a mRNA lacking the 5’-UTR was injected in concert with the UTR morpholino (new Fig. 2H, J). In addition, panel B of this figure represents an in vitro transcription/translation experiment demonstrating the addition of the ATG morpholino blocks translation of the Socs4a protein. The absence of an appropriate antibody precludes other potential analyses.
What is the dark region in Figure 1K? Is is part of the experimental staining or background staining?
- The dark unlabelled region in this panel represents the remaining yolk in this late-stage embryo. This has a distinctive brown color that is distinct from the blue/purple staining of the WISH procedure (be it experimental or background). The Fig. 1 legend text has been modified to provide clarification.
Page 7 line 243: There is probably a typo in Fig. 4K, it is likely to be 4I.
- The typographical error has been corrected.
There is no data that supports a potential interaction between socs4a gene and STAT3 signaling. Thus the conclusions that infer such an interaction should be modified accordingly.
- As suggested, the discussion regarding potential STAT3-socs4a interaction has been revised to make it clear that it remains purely speculative and the mode of regulation unclear.
Round 2
Reviewer 1 Report
The authors have satisfactorily revised the manuscript.
Reviewer 2 Report
The authors have improved the manuscript. It is acceptable in its current form.